# Reinforcement and Deacidification for a Textile Scroll Painting (AD 1881) Using the CNF and MgO Suspensions

**DOI:** 10.3390/polym16070946

**Published:** 2024-03-29

**Authors:** Hanyu Wei, Fangnan Zhao, Yunpeng Qi, Zhihui Jia, Yajun Zhou, Xiaolian Chao, Meirong Shi, Yujia Luo, Huiping Xing

**Affiliations:** 1Engineering Research Center of Historical Cultural Heritage Conservation, Ministry of Education, School of Materials Science and Engineering, Shaanxi Normal University, Xi’an 710119, China; why18821729605@snnu.edu.cn (H.W.); zfn@snnu.edu.cn (F.Z.); qyp@snnu.edu.cn (Y.Q.); jzh1988@snnu.edu.cn (Z.J.); zhouyajun@snnu.edu.cn (Y.Z.); chaoxl@snnu.edu.cn (X.C.); 2Shaanxi Institute for the Preservation of Culture Heritage, Xi’an 710075, China

**Keywords:** the textile scroll painting, conservation treatment, reinforcement and deacidification, nanosized CNF and MgO suspension

## Abstract

The scroll paintings for ancestor trees have been used to inherit the spirit of ancestor worship as a historical record of family development since the late Ming Dynasty in China. A severely degraded scroll painting of an ancestor tree (made of cotton textiles) needs intervention and conservation treatment to mitigate further deterioration. On the basis of the previously reported characterization results for the painting, in this paper, a suspension that is composed of 0.6% cellulose nanofibril (CNF) and nanosized 0.15% MgO in aqueous solvent (denoted as the CNF-MgO susairpension) was prepared. Conventional characterization methods were used to assess the properties of model samples before and after treatment with the CNF-MgO suspension, as well as before and after degradation under two sets of conditions. The results show that the treated model samples are slightly alkaline, given the deposit of alkaline particles, and demonstrate good mechanical properties before and after degradation due to the increase in fiber-to-fiber bond and mitigation of acid-catalyzed hydrolysis. In spite of the non-transparency of CNF and MgO nanoparticles, they have little impact on the optical properties of textiles, as verified by transmittance data and the determination of color changes. This suspension was then used to reinforce and restore the scroll painting in a practical conservation process. The application of CNF and MgO nanoparticles on textile objects investigated in this study would expand our understanding of the conservation of such objects, especially for those that have already become acidic and degraded.

## 1. Introduction

The scroll paintings for ancestor trees have been used to inherit the spirit of ancestor worship as a historical record of family development since the late Ming Dynasty, which has great artistic, cultural, and religious values. A scroll painting for an ancestor tree (drawn in AD 1881 on the textile) needs intervention and conservation treatment due to its severe state of deterioration. The basic properties of this painting have been characterized by our research group, and corresponding results have been published in the study of Zhao et al. [1] (an overview of the painting was presented), demonstrating that the painting support, i.e., the canvas, was made of cotton fibers and painted with mineral pigments. In this study, continuous research was carried out in order to reinforce and deacidify the painting during the process of traditional Chinese conservation treatment.

Reinforcement treatments of textile-based materials can be generally categorized into physical and chemical methods. One of the most commonly used physical methods is the traditional mounting process [2], during which the backing support, made of similar materials as the artifact, is fixed on the back of the textile object [3,4]. The stitching technique is applied to patch holes in the textile objects using threads made of similar materials, and sometimes previous conservation repair materials are removed [5,6,7]. Silk or cotton mesh can also be attached to the textile objects using adhesives, such as PVA-217 aqueous solution [8], on both sides to provide mechanical support. Chemical reinforcement methods mainly involve adding polymeric reinforcement agents, such as polyester-based adhesives [9].

Numerous deacidification treatments have been explored by researchers over decades for cellulosic materials, mainly including aqueous, non-aqueous, and gaseous methods [10,11]. During the aqueous treatment, Ca(OH)_2_, Ca(HCO_3_)_2_, Mg(OH)_2_, Mg(HCO_3_)_2_, and MgO solutions are the most frequently used to neutralize acidic compounds and provide an alkaline reserve for the objects. Water can be substituted for organic solvents such as perfluoro heptane, toluene, methanol, acetone, etc. [12]; for example, C_3_H_6_MgO_4_ in methanol and trifluoro chloroethane for deacidifying textiles was reported [13]. With the development of nanotechnologies, nano- and micro-sized calcium and magnesium-related compounds in both aqueous and non-aqueous media have been introduced by several researchers and heritage institutions [10,14,15]. The gaseous method involves impregnating the paper with gases like morpholine, ammonia gas, cyclohexylamine carbonate, etc. [16].

Based on the brief review of currently available conservation methods for textile objects and the previously reported results, in this study, we optimized the traditional conservation mounting treatment coupled with aqueous reinforcement and deacidification methods to repair the scroll painting, which has become acidic and almost lost fitness-for-use. The treatment is environmentally friendly (without organic solvents) and easy to perform for curators.

Among the reinforcement materials, cellulose nanofibril (CNF) is known to have excellent properties, i.e., high strength and stiffness with a neutral pH. This material has been incorporated into cellulose-based objects, including paper and textile objects [17,18], and has a wide application in papermaking [19,20,21]. CNF is commonly applied to paper objects as wet- and dry-strength additives with high performance to improve paper gloss and barrier properties and for smart and sustainable packaging [18]. It has also been used for the conservation of textile objects in recent years due to its good compatibility, as it retains both crystalline and amorphous regions in cellulose [22,23]. It has been found that CNF could cover and fill in larger pores in the substrate textiles, as well as micropores. We, therefore, aim to explore how CNF affects the properties of the historic textile dating back to the 19th century and to determine the optimal concentration of CNF suspension for enhancing the durability and longevity of this fragile object.

In order to mitigate further acidification and deposit alkaline buffer for the scroll painting, aqueous nanosized MgO suspension was attempted to be used for deacidification due to the advantages of nanoparticle dispersions, e.g., high specific surface area, better penetration and adhesion, etc. [24]. Also, this is due to the fact that the aqueous solvents will not lead to undesirable adhesive solubilization in pigments.

Similar research has been carried out using nano-fibrillated cellulose, combined with CaCO_3_ in an aqueous solvent and MgO in a non-aqueous solvent, in canvas conservation [25]. In this study, it is aimed to investigate the applicability of nanosized MgO suspension on textile objects and the compatibility of nanosized MgO and CNF suspension during traditional conservation treatment to achieve both deacidification and reinforcement effectively.

Prior to the application of nanosized MgO and CNF suspensions for the preservation of the scroll painting, the most appropriate concentration is optimized using model samples made of similar fibers as the painting. A series of basic characterizations were carried out to evaluate the effects of the suspension and stability of the treated model samples, mainly including measurements of pH, color changes, and optical and physical properties. In addition, the nanosized MgO and CNF suspension with optimized concentration was applied to the aged scroll painting for restoration and reinforcement. This study would expand our understanding of conservation for textile objects, especially the objects for which interventional conservation treatment is required.

## 2. Materials and Methods

### 2.1. Samples

The canvas of the scroll painting, made of cotton fibers, was woven using the plain weave method with double warps and single weft, as indicated by the previous results [1], which guided the preparation of model samples in this study.

Before conservation treatment for the scroll painting, model samples were prepared for optimizing concentrations of CNF and MgO suspensions. Given that the elemental analysis for the painting indicates the presence of K, Al, and S, it is speculated that KAl(SO_4_)_2_ might have been applied in the textile as an antiseptic agent or for reinforcing pigments, as suggested by the previous studies [26]. Therefore, model samples were prepared by immersing commercially available cotton textiles in a 1% *w*/*v* KAl(SO_4_)_2_ solution for 10 min and for natural drying, and then the dried samples were degraded under the conditions at 80 °C and 65% RH for 3 days. The degraded samples were stored at room conditions (22 ± 1 °C, 50 ± 10% RH) in the darkness for 48 h before subsequent measurements. The pH of the model samples is 4.5 ± 0.1, consistent with the pH of the scroll painting determined with a flat-tip pH electrode at 4.4 ± 0.1. All chemicals used in this study are provided by Sinopharm Chemical Reagent Co., Ltd. (Shanghai, China).

The reinforcement effect of 0.2–1% CNF suspensions was then prepared and applied to prepare model textile samples, and nanosized 0.01–0.3% MgO suspensions were also prepared to explore the most appropriate concentration for deacidification. The degraded samples were immersed in the suspensions for 10 min to prepare reinforced model samples, which were then used for further characterization.

### 2.2. pH

Paper pH was measured based on the standard cold extraction procedure [27]. A total of 2 ± 0.05 g of samples were cut into 5 mm × 5 mm pieces and placed in 100 mL deionized water in a conical flask with a screw cap. Then, the dispersion in the flask was fully mixed on a vibrator for 2 h, and the supernatants were used for pH measurement using a pH Meter (Mettler Toledo SevenCompact S210, Columbus, OH, USA) with a micro-combined glass electrode (Mettler Toledo Inlab^®^ Micro 51344163, Columbus, OH, USA). The pH of the scroll painting before and after conservation treatment was also determined non-destructively using the flat surface electrode (Mettler Toledo Inlab^®^ Surface Special pH Electrode 51343157, Columbus, OH, USA). Three measurements were carried out for both the model samples and the painting, and the average was used for analysis.

### 2.3. Mechanical Properties

According to the standard [28], the tensile force was determined using 100 mm × 5 mm textile strips with a universal material testing machine (QT-1136PC, Gaotai Testing Instrument Co., Ltd., Dongguan, China), with a tension speed of 100 mm/min. Two sets of sample strips were prepared in both warp and weft directions, and each set consisted of five specimens. Given the good homogeneity of thickness for model samples (thickness uncertainty at 1%, n = 10) prepared in this study, the tensile force was measured and compared before and after treatment, as well as before and after moist and dry heat degradation.

### 2.4. Colorimetry

The CIE *LAB* color space [29], also referred to as the *L**, *a**, and *b** coordinates, stands for lightness from black (0) to white (100) value, green (−) to red (+) value, and blue (−) to yellow (+) value, respectively. It was measured to evaluate color changes. Calibration was carried out using the attached white standard accessory before the spectrophotometer (X-rite VS 450, Grand Rapids, MI, USA) was used. Using the CIE standard illuminant D65 and a 10° observer, measurements were performed 10 times for each model sample [30] before and after treatment with CNF-MgO suspension, and the average was used.

### 2.5. Light Transmittance

In order to investigate the effects of the suspensions on the optical properties of the textile model samples, CNF suspensions and CNF-nanosized MgO suspensions with a series of concentrations were cast onto polystyrene Petri dishes to prepare films to measure light transmittance (%T). A UV/Vis/NIR spectrophotometer (PerkinElmer Lambda 950, Waltham, MA, USA) with a wavelength from 200 to 800 nm.

### 2.6. Accelerated Degradation

In order to explore the stability of model textile samples, the samples before and after the treatment were degraded under two sets of conditions. Firstly, samples underwent dry-heat degradation at 105 °C without controlling RH in the oven. Secondly, they were subjected to moist-heat degradation at 80 °C and 65% RH using an environmental test chamber (Memmert HCP150, Schwabach, Germany) for 3 days. After degradation, the pH, color difference, and mechanical properties were determined.

### 2.7. Traditional Chinese Conservation Treatment

The optimized CNF-MgO suspension was then applied to the scroll painting during the conservation treatment, including steps such as realignment of the distorted threads, cleaning and dust removal, application of CNF-MgO suspension to the canvas, and conservation stitching for patching holes. No extra mechanical support was applied to the scroll painting. To keep the object intact as much as possible, only non-destructive pH measurements were performed for the scroll painting.

## 3. Results and Discussion

### 3.1. Reinforcement for Model Samples

It is known that the application of CNF to paper-based materials has been well studied to largely improve paper mechanical properties [17,18], while little is known about its application to textile objects. The CNF coating on woven and nonwoven fabrics could increase the resistance to air permeability [23], while the reinforcement effect on textiles is rarely studied. We, therefore, attempted to prepare CNF suspensions with a series of concentrations for reinforcing the model samples made of cotton fibers, and the results are shown in Figure 1.

For both strips that are parallel to warp and weft threads, the tensile force increases with increasing concentrations of CNF suspensions at 0.2%, 0.4%, 0.6%, and 0.8%, while starting to decrease when treated with higher concentrations of CNF suspensions at 1%. The mechanical properties of cellulosic materials are mainly determined by fiber length, fiber strength, and fiber-to-fiber bonding areas [18,31]. It is assumed that the amount of the fiber-to-fiber bond in textiles could be increased by introducing the CNF particles, and thus, the mechanical properties can be improved. However, the introduction of nanocellulose can also increase the material stiffness, which might be the reason that tensile force decreased when an excessively high amount of CNF was applied [32].

As shown in Table 1, the pH of treated samples increases from 4.5 to 6.9 with the increasing concentrations of CNF suspensions, which is due to the fact that the pH of CNF is about neutral and the pH values of all samples have slightly decreased after degradation under both conditions. The dry heat condition has a larger effect on pH decrease than the moist heat condition; e.g., for the sample treated with 1% CNF suspension, its pH decreased by 7.2% and 11.6% after dry- and moist-heat degradation, respectively.

The transmittance of CNF films was also explored, showing a decreasing trend with an increase in concentrations of CNF suspensions in the range of the visible spectrum (Figure 2a), as CNF nanoparticles show a white color and are not transparent. However, the transmittance data of all film samples can still be kept at >85%, suggesting a small influence caused by CNF particles on textile transparency. A slight color difference (ΔE) was observed for model samples before and after CNF treatment (Figure 2b). Samples treated with >0.6% CNF suspensions only show a perceivable color difference (2 < ΔE < 4). This finding is consistent with the study by Chinga-Carrasco et al. [33], which found that CNF coatings do not largely affect the optical properties of board materials. The dry heat condition has a greater impact on color changes compared to moist heat degradation for both untreated and treated samples.

Taking into consideration the mechanical properties, pH values, and color changes in model samples before and after treatment with CNF suspensions as explored above, it was found that 0.6% CNF suspension is the most appropriate among all concentrations for reinforcement and will be used for the subsequent experiments. 

The stability of the model textile sample treated with 0.6% CNF suspension was also investigated after dry- and moist-heat degradation, where specimens that are parallel to warp and weft threads were prepared, and the results are shown in Figure 3. The tensile force decreased after degradation for both untreated and treated samples. Also, it is obvious that treated samples all demonstrate higher retention rates of mechanical properties than the untreated, indicating the reinforcing effects of the CNF. Moist heat causes more negative effects than the dry heat condition for our model textile samples.

### 3.2. Deacidification for Model Samples

In order to neutralize acids and create an alkaline buffer in the scroll painting (pH at ~4.5), nanosized MgO in an aqueous solvent was used for deacidification. As reported in the study of Kwiatkowska, Wojech, and Wójciak [34], MgO nanoparticles tend to have better deacidification performance than particles of micrometric size for paper materials. This might be due to the better penetration of nanosized MgO. As shown in Table 2, the pH of model samples treated with 0.6% CNF suspension (denoted as the CNF sample in the subsequent content) shows an increasing trend after treatment with nanosized MgO with a series of concentrations from 0.01 to 0.3%. This indicates that more characterizations are needed to be conducted for selecting appropriate and compatible concentrations of MgO suspension for further deacidification.

In Figure 4, CNF samples present better mechanical properties after treatments with all concentrations of nanosized MgO suspensions, where the tensile force of CNF samples increases with the concentration of MgO suspensions from 0.01 to 0.15% but begins decreasing with the concentration >0.15%, where samples that are parallel to warp and weft threads show the same trend.

Despite the fact that the nanosized MgO is white and non-transparent, this would not be a problem that affects the transparency of treated CNF samples to a large extent. As shown in Figure 5a, the transmittance was explored for the films made of 0.6% CNF and MgO with a series of concentrations. The film samples with ≤0.15% MgO suspensions can keep >90% transmittance, and film samples with 0.15–0.3% MgO suspensions have transmittance >85% in the range of the visible spectrum. Also, color differences in model samples before and after treatment with 0.6% CNF and MgO suspensions are all minimal, with ΔE in the range of 1–2. After degradation under both conditions, ΔE values still stay in an acceptable range (0.5–2).

Based on the above results of pH measurements, mechanical properties, and color changes in the CNF samples treated with these nanosized MgO suspensions, we, therefore, selected the concentration of MgO suspension at 0.15% as the deacidification agent for the scroll painting during the traditional Chinese conservation treatment. Also, the durability of the model sample treated with 0.6% CNF and 0.15% nanosized MgO suspension (denoted as the CNF-MgO sample in the subsequent contents) was explored, as presented in Figure 6. Reinforced and deacidified CNF-MgO samples all show higher retention rates of mechanical properties than the untreated samples after both degradation conditions for both samples that are parallel to warp and weft threads.

### 3.3. Conservation Treatment for the Scroll Painting

Based on the above results, the selected concentration of CNF-MgO suspension was applied to the scroll painting during the process of conservation treatment, as shown in Figure 7a1–a4. Four primary steps include cleaning the painting [6] of distorted and clustering fibers [8], application of the CNF-MgO suspension (the degreasing cotton was used to dip the suspension and was then applied to the painting), and sewing up holes using the cotton thread [2]. The scroll painting after restoration is shown in Figure 7b. It was required to keep the scroll painting intact; only pH measurements were conducted after restoration treatment, which was increased from 4.4 ± 0.1 to 7.2 ± 0.2.

## 4. Conclusions

In order to deacidify and reinforce the textile material, in this study, an appropriate concentration of CNF and nanosized MgO suspension was explored to reinforce and deacidify a scroll painting for the ancestor tree made of cotton fibers, as determined in the previously reported research. Before application of the suspension, model textile samples were prepared, and the pH, optical, and mechanical properties and stability of the model samples were investigated before and after treatment with the suspension.

The mechanical properties of the model samples were improved by depositing CNF particles, which might be due to the fact that CNF particles can improve the amount of the fiber-to-fiber bond in textiles. However, the introduction of nanocellulose can also increase the material stiffness, which might be due to the fact that tensile force decreased when an excessive amount of CNF was introduced. Given the neutral pH of CNF particles, the pH of CNF-treated samples can be slightly improved. Despite the presence of CNF nanoparticles can affect textile transparency, the transmittance data of all treated samples can still be >85%, suggesting a minor influence. Also, slightly perceivable color changes indicate that the introduction of CNF nanoparticles does not largely affect optical properties to a large extent. Considering all these properties, 0.6% CNF suspension is the most appropriate reinforcement and will be used for further explorations.

An investigation of the appropriate concentration of nanosized MgO in an aqueous solvent was carried out. The pH of the CNF samples shows an increasing trend after treatment with 0.01–0.3% nanosized MgO. Also, the treated samples all present better physical properties by testing tensile force, and the sample with the concentration of MgO suspension at 0.15% presents the best performance. In terms of the optical properties of samples before and after treatment with MgO, color changes are within an acceptable range before and after degradation, and samples with ≤0.15% MgO suspensions can keep >90% transmittance. We, therefore, select 0.15% MgO suspension as the deacidification agent for the scroll painting during the traditional Chinese conservation treatment.

Also, the stability of the model sample treated with 0.6% CNF and 0.15% nanosized MgO suspension was explored, showing higher retention rates of mechanical properties than the untreated after degradation for both samples that are parallel to warp and weft threads. This suspension was then applied to the scroll painting for conservation.

## Figures and Tables

**Figure 1 polymers-16-00946-f001:**
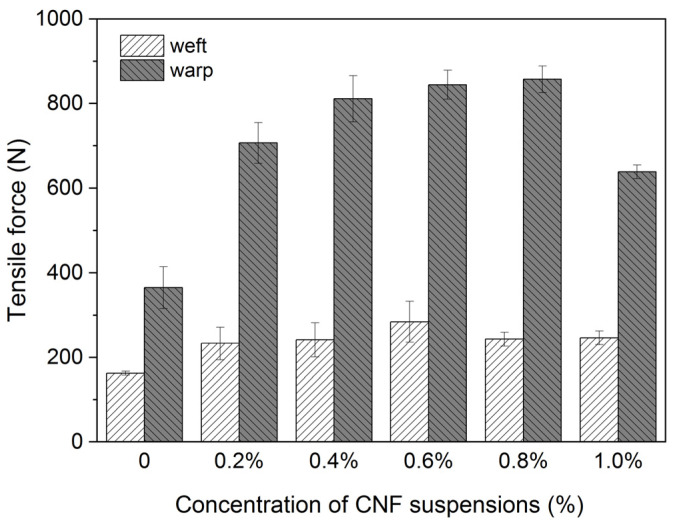
Tensile force of the model textile samples before and after treatment with 0.2%, 0.4%, 0.6%, 0.8%, and 1% CNF suspensions.

**Figure 2 polymers-16-00946-f002:**
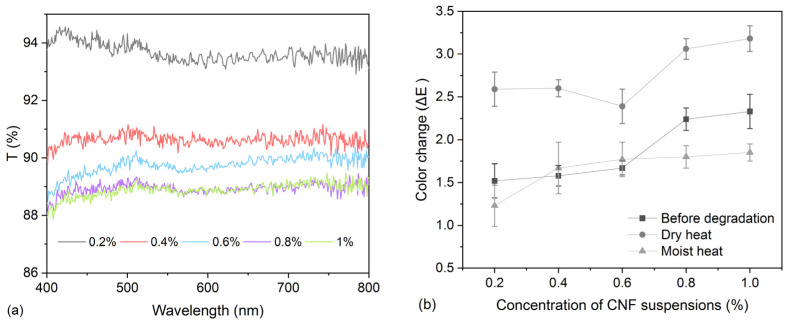
(**a**) Transmittance in the range of the visible spectrum for CNF films with a series of concentrations (from 0.2% to 1%); (**b**) color differences for model textile samples treated with a series of concentrations of CNF suspensions (from 0.2% to 1%) before and after dry- and moist-heat degradation.

**Figure 3 polymers-16-00946-f003:**
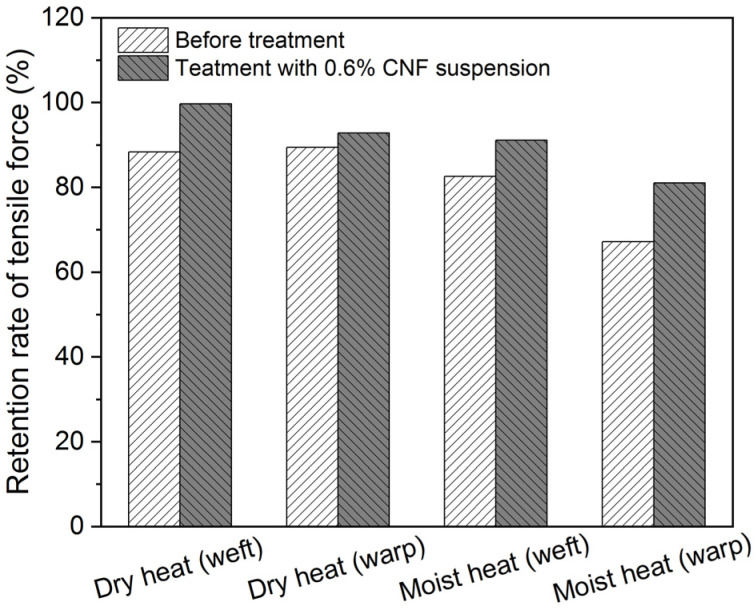
Retention rates of tensile force for model samples before and after treatment with 0.6% CNF suspension after both dry and moist heat degradation.

**Figure 4 polymers-16-00946-f004:**
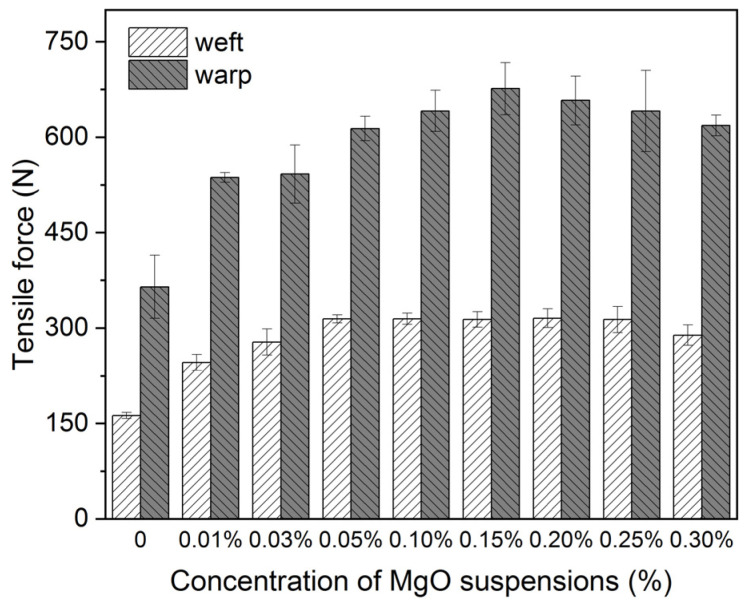
Changes in tensile force for the model textile samples that are parallel to warp and weft before and after treatment with 0.6% CNF suspensions and 0.01%, 0.03%, 0.05%, 0.1%, 0.15%, 0.2%, 0.25%, and 0.3% nanosized MgO.

**Figure 5 polymers-16-00946-f005:**
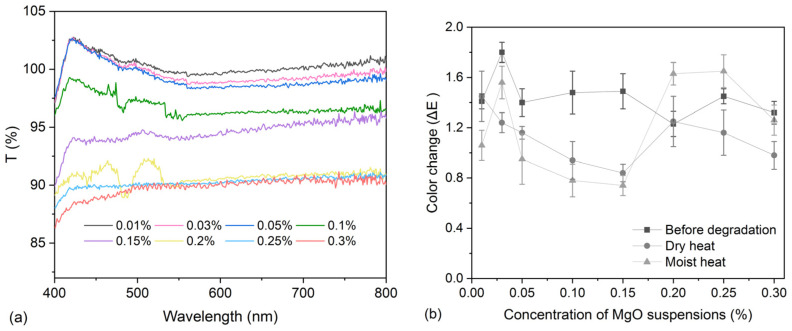
(**a**) Transmittance for films made of a series of concentrations of CNF-nanosized MgO suspensions (from 0.01% to 0.3%) in the range of the visible spectrum and (**b**) color differences for model textile samples treated with 0.6% CNF and a series of concentrations of nanosized MgO suspensions (from 0.01% to 0.3%) before and after dry- and moist-heat degradation.

**Figure 6 polymers-16-00946-f006:**
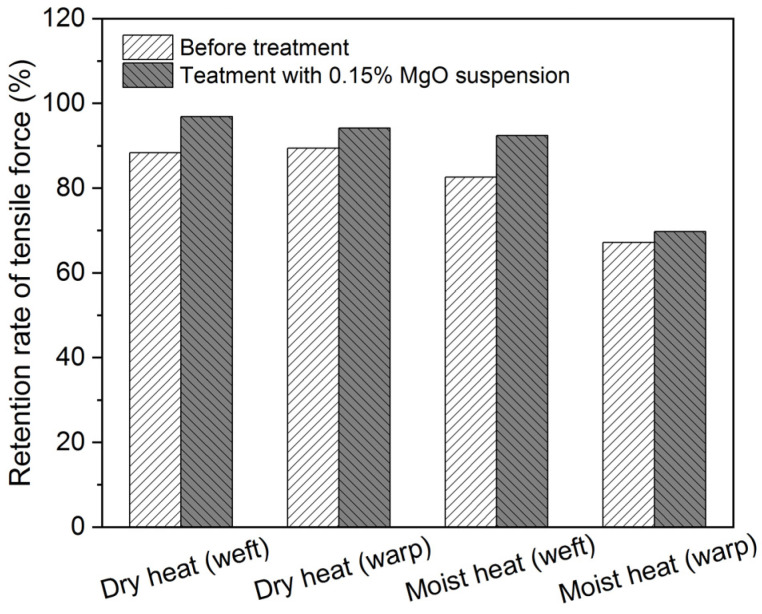
Retention rates of tensile force for model textile samples before and after treatment with 0.6% CNF and 0.15% nanosized MgO suspension after dry- and moist-heat degradation.

**Figure 7 polymers-16-00946-f007:**
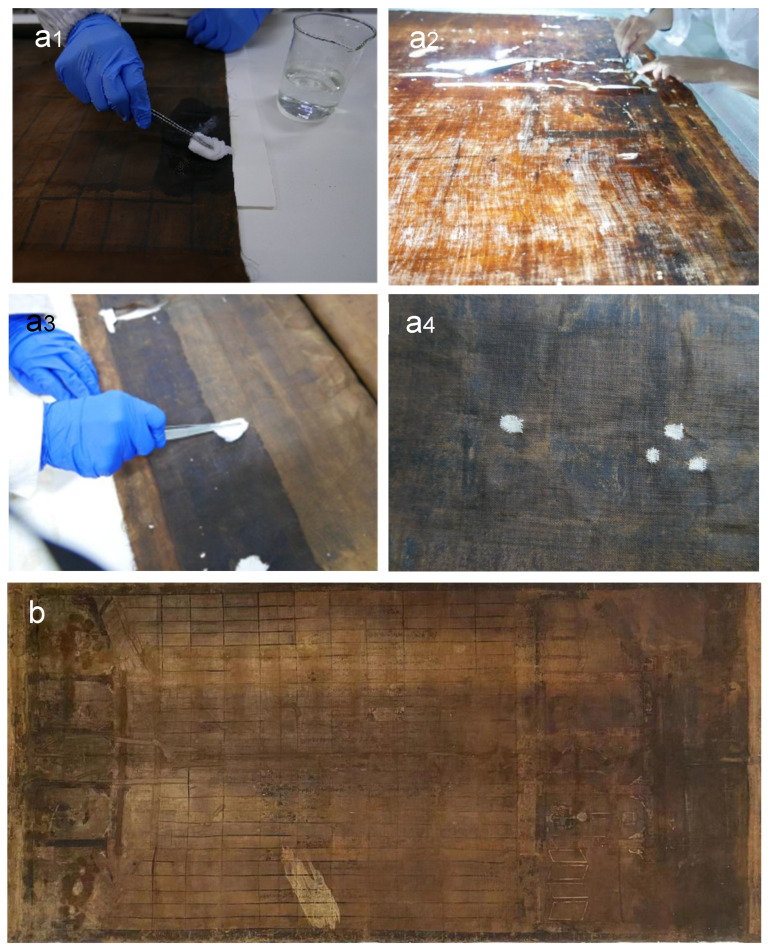
(**a1**–**a4**) Traditional conservation treatment for the scroll painting, including four steps of cleaning, rearrangement of threads, deacidification and reinforcement using CNF-MgO suspensions, and patching holes; (**b**) the scroll painting after conservation treatment.

**Table 1 polymers-16-00946-t001:** pH of model samples before and after treatment with a series of concentrations of CNF suspensions (from 0.2% to 1.2%) before and after moist- and dry-heat degradation.

Treatment Methods		Before Degradation	After Moist Heat	After Dry Heat
Model sample		4.5 ± 0.1	4.4 ± 0.1	4.4 ± 0.1
pH of model samples treated CNF suspensions	0.2%	5.6 ± 0.1	5.9 ± 0.2	5.6 ± 0.1
	0.4%	5.6 ± 0.2	6.3 ± 0.2	5.6 ± 0.2
	0.6%	5.9 ± 0.3	6.3 ± 0.2	5.9 ± 0.3
	0.8%	6.0 ± 0.2	6.4 ± 0.3	6.0 ± 0.2
	1%	6.1 ± 0.3	6.4 ± 0.2	6.1 ± 0.3

**Table 2 polymers-16-00946-t002:** pH of model samples treated with 0.6% CNF suspension and nanosized MgO with a series of concentrations from 0.01% to 0.3%, in which the CNF sample represents the model sample treated with 0.6% CNF suspension.

Treatment Methods		Before Degradation	After Moist Heat	After Dry Heat
CNF sample		6.5 ± 0.3	6.3 ± 0.2	5.9 ± 0.3
The CNF sample treated with a series of concentrations of nanosized MgO suspensions	0.01% MgO	7.4 ± 0.2	6.8 ± 0.2	6.5 ± 0.1
	0.03% MgO	8.0 ± 0.1	7.6 ± 0.2	7.2 ± 0.2
	0.05% MgO	8.4 ± 0.2	8.2 ± 0.1	8.2 ± 0.2
	0.1% MgO	8.7 ± 0.1	8.7 ± 0.3	8.6 ± 0.2
	0.15% MgO	9.4 ± 0.1	9.1 ± 0.2	9.2 ± 0.2
	0.2% MgO	9.9 ± 0.2	9.6 ± 0.1	9.4 ± 0.3
	0.25% MgO	9.8 ± 0.2	9.7 ± 0.2	9.6 ± 0.1
	0.3% MgO	9.7 ± 0.2	9.6 ± 0.1	9.2 ± 0.1

## Data Availability

Data are contained within the article.

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
