# Peer review of "Reinforcement and Deacidification for a Textile Scroll Painting (AD 1881) Using the CNF and MgO Suspensions"

_polymers, 2024, doi:10.3390/polym16070946_

Round 1
Reviewer 1 Report
Comments and Suggestions for Authors
The paper entitled Reinforcement and deacidification for a textile scroll painting 2 (AD 1881) using the CNF and MgO suspensions explores a mix suspension to treat compromised textile. With the help of new specimens, the authors evaluate the concentration of CNF and MgO based on aging using a multi-analytical approach. In particular, mechanical tests, pH, and possible color changes are explored. The experiment is well-designed, and the data is correlated; however, some minor aspects must be revised and corrected. In the attached documents, the authors can find my comments. I would also like to suggest the authors revise the English as the structure of a few sentences is quite complex—shorter sentences avoiding repetitions might help. I would also invite the authors to comment better on the suspension application to the ancient scroll painting. Details on the applications (brush? Spray?) and supporting data (pH?) might help verify the treatment's effect.

A few sentences must be revised as too long and full of repetitions
Author Response
A few sentences must be revised as too long and full of repetitions
Response: Thank you and English has been polished in the text.
do you have any idea about the density of the textile? this might effect the products response in comparison to real artwork
Response: Thanks for your question. The textile used for prepared model samples are commercially available, made of pure cotton fibers.
do you have any data to support this? Personally, 3 days even at 80°C might won't get any significant degradation. can you comment on this? the ISO you cite is referred to papar and board notto textile. Why do you think you can apply on textile?
Response: Thank you very much for your comments. Indeed, the degradation of cellulosic samples before acidification at 80°C for 3 days won’t get significant results. However, in our study, it is the sample treated with KAl(SO4)2 was used, where KAl(SO4)2 is acidic.
Thank you, the ISO standard for the degradation of paper materials was used as the reference, as both paper and textiles are cellulosic materials. This has been revised in the manuscript.
please add if you used SCI or SCE component and how may indipendent points were considered and averaged
Response: Thank you very much for you comments. In this study,
SCI and SCE are generally only found in d/8 construction colorimeters and are the two modes of color measurement. In our study, the colorimeter we used is 45°/0° spectrophotometer, where SCI and SCE cannot be selected, but the color detected using this way is similar to the SCE mode. As indicated in this section, measurements were performed for 10 times for each model sample before and after treatment with CNF-MgO suspension, and the average was used.
I am sorry but between the 0.6 amnd 0.8% the dicrease is not so evident and considering the associated errors the values are almost comparable for the warp
can you better comments on this
Response: Thank you for your comments. This is just one of the experimental results about tensile force, and other characterization results should also be considered to confirm 0.6% presents better performance.
the data you report in the comments do not match with the table
please revise it
Response: Thank you comments. This sentence has been revised in the manuscript.
please add in figure 2 the data ralted to the untreated sample as a reference to better read the influence of the treatment
Response: Thank you for your comments. We may not well clarify Figure 2a, the transmittance of CNF films was explored and presented in Fig. 2a, rather than the model samples treated with CNF.
why do you think that in particular the color change is lower in the case of the dry heat ?
can you comment on this?
respect the others it is possible to see an increasing trend with increasing CNF concentration
Response: As shown in Figure 2b, for model samples treated with different concentrations of CNF, color difference of the samples before and after dry-heat is larger than that of the textile samples before and after moist-heat.
as already commented before you should add values referred to untreated samples that were subjected to the same againg in order also to better understand if the treatment might reduce possible color changes
Response: Many thanks for spotting this out. In Figure 5a, it is the transmittance of CNF-MgO films rather than that of the textile samples.
for conservators it will be useful to add more information on how the application of suspention was done on the textile. comments on the visual aspect and materials after treatments should be added
Response: Many thanks and this has been modified in the manuscript.
Given that the scroll painting is a heritage object, we’ve only determined pH for the painting before and after restoration, where pH increased from 4.4 to 7.2. Other characterization was not carried out in this study.
do you have any data such as pH measurments before and after treatment to ferify the succesful deacidification treatments?
Response: Thank you and this has been added to the manuscript.
this aspect was not verified in the ancient samples and you cannot give this as done just based on the specimens at least a pH data should be added
Response: Thank you and this has been revised in the manuscript.
Reviewer 2 Report
Comments and Suggestions for Authors
Manuscript ID: polymers-2913440
The manuscript is devoted to the elaboration of methods for reinforcement and deacidification of old textile scroll paintings using cellulose nanofibers and MgO suspensions. The manuscript is interesting and the subject is definitely worth pursuing. However, I think that the paper would be much better suited for a materials chemistry oriented journal e.g. materials MDPI since it has very little to do with polymers. Of course, cellulose is a polymer but CNF should be rather regarded as a kind of nanoscale material.
Moreover, the following is not clear to me:
1. Fig.1 – have you studied 2% of CNF to confirm the observed trend? Can the result with 1% of CNF be accidental? Why the trend observed for weft is different to that observed for warp?
2. Lines 187-190 – why addition of 0.2% of neutral CNF caused that large increase of pH? Why addition of larger amounts of CNF did not cause larger increase of pH than that recorded?
3. Lines 206-207 and 223-224 – please discuss and explain the observed phenomena. What can be the reason for the opposite trends?
4. Figure 4 – have you studied the effect of MgO with other concentrations of CNF? Is the trend presented in Fig.4 general?
Comments on the Quality of English LanguageEnglish – some minor corrections should be made.
Please remove lines 163-165
The first sentence of Conclusions should be corrected.
Author Response
- 1 – have you studied 2% of CNF to confirm the observed trend? Can the result with 1% of CNF be accidental? Why the trend observed for weft is different to that observed for warp?
Response: Thank you very much for your question, the reinforcement effect of 0.2 – 1% CNF suspensions was prepared and applied in model textile samples to observe the trend. A slow increase in tensile force can be seen with up to 0.8% concentration.
As introduced in the first paragraph of “Samples”, the canvas of the painting was woven using the plain weave methods with double warps and single weft, as indicated in the previous research. We therefore selected the cotton textile made of thicker warp to prepare model samples.
- Lines 187-190 – why addition of 0.2% of neutral CNF caused that large increase of pH? Why addition of larger amounts of CNF did not cause larger increase of pH than that recorded?
Response: Many thanks for your question. As indicated in Line 187, given that pH of CNF is about neutral, for model samples treated with CNF, their pH values tend to be neutral.
- Lines 206-207 and 223-224 – please discuss and explain the observed phenomena. What can be the reason for the opposite trends?
Response: In order to explore the stability of model textile samples, the samples before and after the treatment were degraded under two sets of conditions at 105 oC (dry-heat degradation without setting RH) using the oven and at 80 oC, 65 % RH (moist-heat degradation). The dry-heat condition caused larger effects on the stability of model samples, as the results presented in Line 206-207, and 223-224. This is due to that higher temperature leads to higher degradation rates.
- Figure 4 – have you studied the effect of MgO with other concentrations of CNF? Is the trend presented in Fig.4 general?
Response: Thank you very much for your question. In this study, we first selected the most appropriate concentration of CNF for reinforcing degraded textile samples as used in this study, which is 0.6%, as explained in the Lines 209-212. However, whether this trend is applicable in other cases, such as other types of samples, should be further explored.
English – some minor corrections should be made.
Response: Many thanks for your comments and English has been further polished.
Please remove lines 163-165
Response: Thank you very much for spotting this out. Lines 163-165 have been removed in the manuscript.
The first sentence of Conclusions should be corrected.
Response: Thank you for your suggestion and the first sentence of Conclusions has been removed.
